# Low-Volume Nodal Metastasis in Endometrial Cancer: Risk Factors and Prognostic Significance

**DOI:** 10.3390/jcm9061999

**Published:** 2020-06-25

**Authors:** Virginia García Pineda, Alicia Hernández Gutiérrez, Myriam Gracia Segovia, Jaime Siegrist Ridruejo, María Dolores Diestro Tejeda, Ignacio Zapardiel

**Affiliations:** Gynecologic Oncology Unit, La Paz University Hospital—IdiPAZ, 28046 Madrid, Spain; aliciahernandezg@gmail.com (A.H.G.); dra_gracia@hotmail.com (M.G.S.); jaimesiegrist@hotmail.com (J.S.R.); mdtejeda@gmail.com (M.D.D.T.); Ignaciozapardiel@hotmail.com (I.Z.)

**Keywords:** low-volume metastasis, ultrastaging, endometrial cancer, sentinel node biopsy

## Abstract

**Objective:** To evaluate the oncological outcomes of patients with low-volume metastasis compared to those with macrometastasis and negative nodes in endometrial cancer. **Methods:** A single institutional retrospective study was carried out, which included all patients with endometrial cancer who underwent surgical treatment between January 2007 and December 2016. We analyzed the progression-free survival (PFS) and overall survival (OS) of all patients after sentinel node biopsy and full nodal surgical staging according to their final pathological nodal status, focusing on the impact of the size of nodal metastasis. **Results:** A total of 270 patients were operated on during the study period; among them, 230 (85.2%) patients underwent nodal staging. On final pathology, 196 (85.2%) patients had negative lymph nodes; low-volume metastasis (LVM) was present in 14 (6.1%) patients: 6 (2.6%) patients had isolated tumor cells (ITCs) and 8 (3.5%) patients presented just micrometastasis; additionally, 20 (8.7%) patients presented macrometastasis. After a median (range) follow-up of 60 (0–146) months, patients with macrometastasis showed a significantly worse PFS compared to LVM and node-negative patients (61.1% vs. 71.4% vs. 83.2%, respectively; *p* = 0.018), and similar results were obtained for 5-year OS (50% vs. 78.6% vs. 81.5%, respectively; *p* < 0.001). Half of the patients presenting LVM did not receive adjuvant treatment. Moreover, LVM had a moderate nonsignificant decrease in 5-year PFS compared to node-negative patients. **Conclusions:** Patients with endometrial cancer and low-volume nodal metastasis demonstrated a better prognosis than those presenting macrometastasis. Low-volume metastasis did not show worse oncological outcomes than node-negative patients, although there was a slight decrease in progression-free survival.

## 1. Introduction

Endometrial cancer (EC) is the most common gynecological malignancy in developed countries, with an estimated 65,620 new cases in 2020, causing 12,590 deaths annually in the USA. Globally, 382,069 new cases of EC were diagnosed in 2018, with 89,909 deaths worldwide [1,2].

Classic management of early-stage EC is based on hysterectomy, bilateral salpingo-ophorectomy (BSO), and depending on the pathological characteristics, pelvic and para-aortic lymph node dissection (LND) from GOG 33 trial results published in 1988 [3]. Nowadays, the evaluation of lymphatic node status in endometrial cancer is still recommended by many Gynecological Oncology Societies to tailor the appropriate treatment. Since lymph node status represents one of the most important prognosis factors, in 2015, the last consensus guideline published by the American College of Obstetricians and Gynecologist and the American Society of Gynecologic Oncology (SGO) recommended the importance of retroperitoneal staging at initial management [4].

Systematic lymphadenectomy in the surgical management of EC remains controversial since two randomized trials failed to demonstrate the therapeutic role of this procedure, with no additional improvement in terms of overall or recurrence-free survival and showing a morbidity increase due to lymphocele formation, lymphedema, and prolonged operative time [5,6].

Over the last decade, several publications have shown that the use of sentinel lymph node (SLN) mapping is an effective alternative for patients with presumed early-stage EC. The most recent guidelines published by the National Comprehensive Cancer Network established that the performance of SLN mapping in EC can be considered for the surgical staging of apparent uterine-confined malignancies [7]. Several groups have published their results with the SLN biopsy technique in EC, reporting high sensitivity and low false-negative rates, as well as good negative predictive values [8,9,10]. In fact, a recent prospective clinical trial has demonstrated its safety when used for surgical staging purposes in early-stage high-risk EC [11].

There are two main advantages to SLN mapping in EC. First, an SLN biopsy showed a morbidity decrease compared to complete lymphadenectomy in terms of lymphedema and lymphocele formation. Second, the ultrastaging of SLNs showed higher sensitivity to detecting nodal metastasis, mainly due to low-volume metastasis (LVM) detection [12]. LVM could represent 25% to 63% of all positive SLNs, which means an 8% increase in nodal positivity compared to regular pathologic staging [13]. However, the therapeutic implications and prognostic value of LVM detection still remain controversial, and there is a gap in the current adjuvant management of EC in these cases.

The main aim of our study is to evaluate the oncological outcomes of patients with low-volume metastasis compared to those with macrometastasis and negative nodes in endometrial cancer.

## 2. Experimental Section

This is a retrospective single-institutional study that collected patients with endometrial cancer treated surgically at our Institution from January 2007 to December 2016. Patients with primary sarcoma or metastatic disease at diagnosis were excluded. Institutional review board approval was obtained before conducting the study, with reference number PI-3846.

The patients with a low risk of recurrence (FIGO stage IAG1-G2) underwent a total hysterectomy, BSO and SLN technique, or pelvic lymphadenectomy (PLND). Patients with intermediate-risk (FIGO stages IBG1-G2 or IAG3) or high-risk patients (FIGO stages IBG3 or II and endometrioid tumors, and also FIGO stages I–II with serous carcinoma, carcinosarcoma, or clear cell carcinoma) also underwent para-aortic LND and total omentectomy (if nonendometrioid histology). Most of the surgeries were conducted by laparoscopy. Sentinel lymph node protocol was carried out in all cases from its introduction in January 2012 (it was the only nodal-staging procedure for low-risk patients).

### 2.1. Sentinel Lymph Node Protocol

We used the protocol published by Delpech et al. in 2007, with the dual tracer technique [14]. Firstly, two cervical injections at 3 and 9 o’clock (5 mm superficial and 15 mm deep) of 2 mL of technetium sulfur colloid were administered with a 25-gauge spinal needle the day before surgery. Lymphoscintigraphy images were obtained 2 h after the injections, and from then on, every 30 min until SLN visualization with the integration of single-photon emission computed tomography (SPECT/CT) [15,16]. Intraoperatively, 4 mL of patent blue (methylene blue or isosulfan blue) or indocyanine green dilution 2.5 mg/mL (ICG) were injected cervically (2 mL per spot, 5 mm superficial and 15 mm deep) [17]. At the beginning of the surgery, all the pelvic areas were carefully inspected for lymph ducts, following the main lymphatic drainage pathways (upper paracervical pathway, lower paracervical pathway, and infundibulopelvic pathway). Lymph nodes marked by technetium (hot lymph nodes) and/or those marked by ICG were selected and removed.

### 2.2. Histopathological Evaluation

The sentinel lymph node pathologic ultrastaging was performed with multiple sectioning at 200 µm intervals. Each section was also sectioned at 50-µm intervals and stained with hematoxylin and eosin. An additional slide of each interval was used for an immunohistochemistry exam (IHC) with an anticytokeratin antibody dilution (cytokeratins AE1–AE3).

Lymph node status was defined using the criteria of American Joint Committee on Cancer for breast cancer (2002): Isolated tumor cells (ITCs) were defined as a focus of metastatic disease measuring ≤0.2 mm, micrometastasis (MIC) was defined as a focus of metastatic disease between 0.2 and 2 mm, and macrometastasis (MAC) was defined as a focus of metastatic disease >2 mm [18]. Those lymph nodes without tumors present on pathologic evaluation were reported as negative. LVM is defined as ITCs and MIC together.

### 2.3. Statistical Analysis

Qualitative variables were reported with absolute numbers and percentages. Quantitative variables were reported as median and range. Categorical variables were compared using the chi-square test for univariate analysis. A multivariate analysis was carried out using logistic regression. Recurrence-free survival (PFS) and overall survival (OS) were calculated by the Kaplan –Meier method and the Mantel–Cox statistical test. The alpha error was set at 5%. All statistical analyses were performed using SPSS Statistics v.24.0 (IBM Corp., Armonk, NY, USA).

## 3. Results

A total of 270 patients diagnosed with FIGO Stages I–II of endometrial cancer, who underwent surgery, were reviewed in this study. Demographic and final clinicopathological features and the surgical approach of our study population are summarized in Table 1.

Lymph node staging was performed in only 230 (85.2%) patients. Among them, pelvic and para-aortic lymphadenectomy (PAL) was performed in 187 (69.2%) cases and sentinel lymph node biopsy in 106 (39.3%) patients. Para-aortic lymphadenectomy by laparoscopy was performed in 79 (29.3%) patients, most of them by the retroperitoneal approach (75.9% of all laparoscopic PAL).

For the entire study population who received nodal staging with SLNB and/or PLND ± PAL, a total of 314 SLNs and 4275 non-SLNs were removed. A median (range) of 2.6 (0–9) sentinel lymph nodes, 14 (0–36) pelvic lymph nodes, and 17 (2–39) paraaortic lymph nodes were harvested per patient.

On final pathology, 196 (85.2%) patients had negative lymph nodes. LVM was present in 14 (6.1%) patients. Among them, 6 (2.6%) patients presented ITCs and 8 (3.5%) patients presented MIC. Finally, 20 (8.7%) patients presented macrometastasis. Nine (64.3%) cases of LVM were detected only at SLN evaluation, thanks to the pathological ultrastaging (5 cases of ITCs and 4 cases of MIC). Among the positive nodes, 34 (10.8%) and 14 (41.2%) were LVM, and the remaining 20 (58.8%) were MAC. The final FIGO stage was distributed, as reported in Table 1.

Uterine risk factors for nodal metastasis, ≥50% of myometrial invasion, and lymphovascular invasion were evaluated in our population *(*Table 1*).* The comparison of the different risk factors between the types of nodal metastasis is shown in Table 2. The only significant independent factor in the multivariate analysis for nodal involvement was myometrial invasion ≥50% with a significant OR = 2.9 (CI95% 1.1–7.7; *p* = 0.032). In univariate analysis, ≥50% myometrial invasion and the presence of lymphovascular invasion were significantly higher in patients with MAC and LVM compared to those who presented negative nodes.

Adjuvant therapy was significantly higher in the group of MAC and LVM compared to patients with negative nodes (*p* < 0.001). Nevertheless, 6 (42.9%) patients with LVM received chemotherapy in contrast with 15 (83.3%) patients with MAC. Detailed information on the type of adjuvant treatment is reported in Table 3.

Patients underwent a median (range) follow-up of 60 (0–146) months. They showed a PFS of 82.4% and an OS of 78%. When we grouped them according to nodal status, we found 5-year PFS and OS of 61.1% and 50% for MAC, 71.4% and 78.6% for LVM, and 83.2% and 81.5% for negative nodes, respectively, with significant differences among groups (*p* = 0.018 and *p* < 0.001; respectively; Figure 1 and Figure 2).

Among the LVM cohort, we observed four recurrences, two cases with ITCs, and two with MIC, all of them distant metastasis. Among them, 3 (75%) patients presented endometrioid histology, with lymphovascular invasion and myometrial invasion ≥50% or cervical stromal involvement. All data of LVM patients are detailed in Table 4.

## 4. Discussion

In our population of 230 patients with lymph node staging, we evaluated the oncological outcome, in terms of PFS and OS, in those patients with LVM comparing with those with negative or MAC in the lymph nodes retrieved. Nowadays, the diagnosis of LVM in endometrial cancer remains a challenge for Oncological Societies to tailor adjuvant treatment, as its prognosis is not yet well established. As SLNB was incorporated into our institutional protocol of EC management in 2012, for the patients treated between January 2007 and December 2011, SLNB was not routinely performed, even though it was carried out on 106 patients. NCCN guidelines have recognized SLNB as an acceptable alternative to complete lymphadenectomy in the early stages of EC since 2014, and recently, it has extended the indication to early high-risk EC. Conversely, the ESMO-ESGO-ESTRO Consensus Conference on EC [19] still recommends the application of SLNB only in the context of controlled trials; however, its last update was in 2016, and additional scientific evidence has been published since then [8,10].

The most important prognostic factor that influences overall survival in patients with endometrial cancer is lymph node involvement, which is decisive to tailor adjuvant treatment. The addition of sentinel lymph node mapping in patients with apparent uterine-confined endometrial cancer has a significant impact on the detection of metastases at the expense of LVM. The study of Holloway et al. [12] included 780 patients with endometrial cancer, where the group of sentinel node mapping had twice as many lymph node metastases as the nonmapped group (30.3% vs. 14.7%, *p* < 0.001), the majority due to the identification of micrometastases and ITCs. The high detection of low volume metastases in this study was associated with increased use of adjuvant therapy.

The SENTI-ENDO study reported promising results concerning SLN ultrastaging, as 47% of metastatic SLNs were diagnosed by immunohistochemistry and not with conventional histology. In the same way as this study, SLN ultrastaging changed the staging of 11% of low-risk cases and 15% of intermediate-risk cases to a higher risk, respectively [9,20]. More recent trials like FIRES and FILM studies have reported similar results. The FIRES prospective trial by Rossi et al. concluded that nodal metastases were identified in the sentinel lymph nodes in 97% of patients with positive nodal status, yielding a sensitivity to detect node-positive disease of 97.2% (CI95% 85–100%), and a negative predictive value of 99.6%. Moreover, 54% of all positive SLNs were LVM, and all of them were diagnosed by just an immunochemistry exam. The FILM trial by Frumovitz et al. reported similar rates; 62% of LVM was detected only in SLNs by ultrastaging [8,10]. In our cohort with positive lymph nodes (34 cases), LVM was detected in 14 cases (41.2%), where 64.3% were detected only by SLN ultrastaging, according to reported results in the literature. Although it is clear that SLNB improves the detection of nodal disease, the prognostic role of LVM is still unclear.

In this study, patients with macrometastasis received significantly more adjuvant therapy based on chemotherapy and external beam radiotherapy (EBRT) compared with LVM and negative lymph node cohorts (83.3% vs. 42.9% vs. 12.1% respectively, *p* < 0.001; Table 3*).* Nevertheless, within the LVM cohort, three (21.4%) patients with ITCs received chemotherapy ± EBRT and one (7.14%) patient received EBRT ± BT based on high-risk uterine features. On the other hand, within the micrometastasis cohort, three (21.4%) patients received adjuvant therapy with chemotherapy ± EBRT and three (21.4%) patients received EBRT ± BT for high-risk uterine characteristics.

Despite the differences in adjuvant treatments administered to patients according to nodal status, the oncological outcomes of patients with LVM or negative lymph nodes were better than those of patients who presented macrometastasis. The OS within the macrometastasis cohort was statistically worse, with a median follow up of 60 months (0–146; 50% MAC vs. 78.6% LVM vs. 81.5% negative nodes, *p* < 0.001) in accordance with published literature (Figure 1) [21]. Plante et al. [21] published a study about the prognosis of ITCs in EC, which reported a median follow-up of 29 months of PFS in the ITC cohort of 95.5%, in micrometastasis patients 85.5%, similar to negative nodes (87.6%) and statistically better than patients with macrometastasis (58.5%; *p* = 0.0012). Further, patients with macrometastasis had significantly poorer survival than patients with ITCs (58.6% vs. 95.5%).

In our series, the results were similar. PFS with a median follow-up time of 60 (0–146) months was also significantly worse in the macrometastasis group (61.1%) compared to the LVM group (71.4%) and negative lymph nodes (83.2%) (*p* < 0.05; Figure 2). Clair et al. [22] reported similar results. They published a retrospective series of 800 patients with EC treated surgically, reporting a cohort of 44 cases of LVM (5.2% of the entire population). In contrast with our results, they reported no differences between adjuvant treatments administered to the three different groups. With a median follow-up of 26 months, the 3-year PFS for the negative nodes group was 90%, for LVM was 86%, and for macrometastasis was 71% (*p* < 0.001). They concluded that adjuvant therapy based on chemotherapy improves the survival rates in patients with LVM compared to patients with macrometastasis. Concordant results were described by Ignatov et al. [23] in a recent publication just for micrometastasis; with adjuvant therapy, the median PFS of patients with nodal micrometastases was similar with those of node-negative patients (*p* = 0.648). Conversely, without adjuvant therapy, the PFS in the cohort of patients with micrometastases was significantly smaller in comparison to PFS in the node-negative cohort (*p* = 0.0001).

Most of the recently published series about prognosis significance of LVM established similar outcomes in PFS and OS between groups of ITCs/micrometastasis when patients received adjuvant treatment compared with the negative-nodes group [22,23,24]. These results are not concordant with ours, as we did not find significant differences in 5-year PFS between LVM and negative nodes, in spite of the fact that less than 50% of patients with LVM did not receive chemotherapy.

We could not discriminate between ITCs and micrometastasis due to our LVM sample size, but PFS after a median follow-up of 60 months seemed to be lower compared with the negative-nodes group (71.4% vs. 83.2%), although no statistical difference was found.

We reported that within the LVM cohort, four patients experienced recurrences (28.6%), including two cases with ITCs and two with micrometastasis, all of them metastatic recurrences. Most of these patients (75%) presented high-risk uterine factors and received adjuvant therapy. Our recurrence rate was also concordant with the literature. Todo et al. reported [25] that 28.6% of patients with LVM who received adjuvant therapy recurred. Moreover, they found worse histopathological features in these patients than in the negative-nodes group. They described that the presence of LVM was an independent risk factor for recurrence. The 8-year OS and PFS rates were more than 20% lower in the LVM group than in the node-negative group (OS: 71.4% vs. 91.9%; RFS: 55.6% vs. 84.0%); however, this was not statistically significant.

Regarding histopathological uterine risk factors associated with all different nodal metastasis sizes, we concluded that most of the LVM cases had endometrioid histology and low grade, but there were no statistical differences between macrometastasis and negative lymph node cohorts. Conversely, patients with macrometastasis presented more often ≥50% myometrial invasion or cervical stromal involvement and lymphovascular invasion than the other two groups. Clinton et al. retrospectively studied the histopathological features of patients with endometrial cancer associated with lymph node LVM; their results were consistent with ours concerning histopathological grade—all patients were significantly more likely to have Grade 1 endometrioid carcinoma with less than 50% of myometrial invasion. However, in our LVM cohort, 50% of patients presented ≥50% myometrial invasion or cervical stromal involvement. Myometrial invasion was the only histopathological characteristic that behaved like an independent risk factor for lymph node disease in our study population [26].

In this study, we report no differences between the overall survival of patients presenting LVM and patients with node-negative diagnosis, despite that over 50% of LVM did not receive adjuvant treatment. Additionally, patients diagnosed with macrometastasis were treated with chemotherapy in over 80% of cases and showed a significantly lower survival rate than the LVM group.

However, a higher recurrence rate was observed in the LVM cohort when compared to the node-negative patients. Nevertheless, we must consider that the majority of these recurrences appeared in patients with uterine features that confer a high risk of recurrence.

According to our results, patients in whom ITCs or micrometastasis were detected would be treated by tumor characteristics, which is consistent with reported literature [27], although a recent review of literature about the prognostic role of ITCs/micrometastasis established that LVM had an increased relative risk of recurrence compared to negative patients, despite adjuvant treatment is being administered [28].

## 5. Conclusions

In conclusion, patients with LVM have significantly better survival than patients with macrometastasis. LVM has not been shown to be a worse prognostic factor than negative nodes, although worse progression-free survival has been observed, with nonsignificant differences. However, given the infrequency of LVM presence, just a few series have been published in the literature evaluating the impact of LVM at node level in endometrial cancer, considering that it may negatively influence the recurrence rate. The use of sentinel node biopsy in nodal staging in endometrial cancer improves LVM detection thanks to the ultrastaging evaluation, and it should be implemented in current practice. Multi-institutional prospective trials would be required to assess the real prognostic value of LVM and to tailor the adjuvant treatment accordingly.

## Figures and Tables

**Figure 1 jcm-09-01999-f001:**
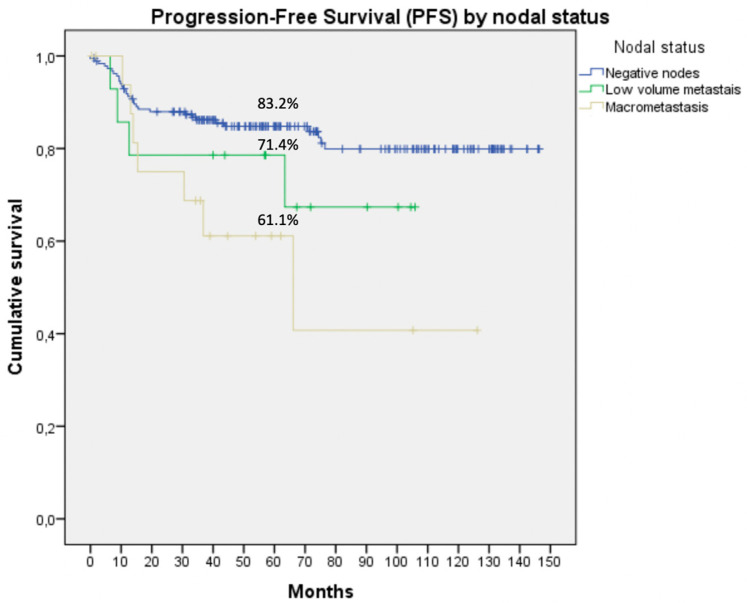
Progression-free survival by nodal status in patients with nodal surgical staging. Significant differences were observed between groups in 5-year PFS (*p* = 0.018).

**Figure 2 jcm-09-01999-f002:**
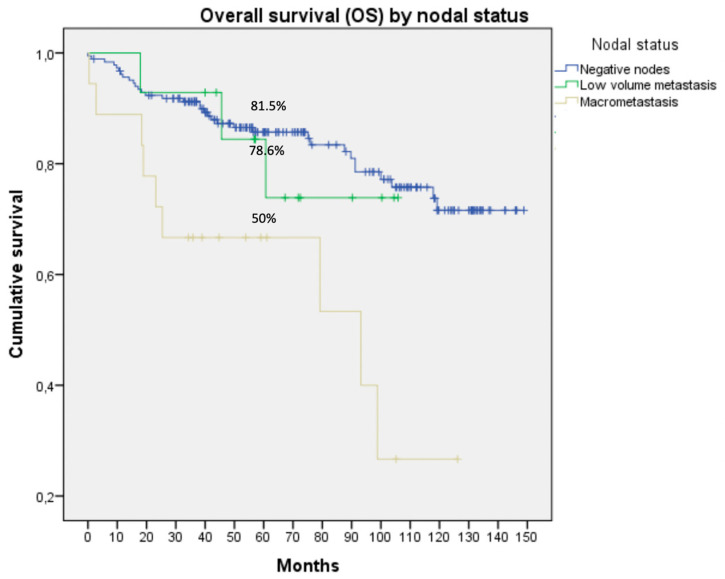
Overall survival by nodal status for the cohort with nodal staging. Significant differences were observed between groups in 5-year OS (*p* < 0.001).

**Table 1 jcm-09-01999-t001:** Patients and tumor characteristics (*n* = 271).

Variables	N (%)
**Age (years)** Median (range)	65 (32–89)
**BMI** Median (range)	29 (17–50)
**Surgical approach**	
Laparotomy	27 (10.1%)
Laparoscopy	236 (88.4%)
Vaginal	4 (1.5%)
**Histology**	
Endometrioid	228 (84.1%)
Serous carcinoma	21(7.7%)
Clear cell carcinoma	15(5.5%)
Carcinosarcoma	6 (2.2%)
Other	1 (0.4%)
**Grade**	
1	145 (53.7%)
2	51 (18.9%)
3	74 (27.4%)
**Lymph-vascular invasion**	
Negative	199 (74.3%)
Positive	69 (25.7%)
**Myometrial invasion**	
< 50%	174 (64.2%)
≥ 50%	97 (35.8%)
**FIGO Stage**	
IA	164 (46.5%)
IB	57 (21%)
II	8 (3%)
IIIA	9 (3.3%)
IIIB	3 (1.1%)
IIIC1	20 (7.4%)
IIIC2	8 (3%)
IVA	1 (0.4%)
IVB	1 (0.4%)
**Lymph node status (*n* = 230)**	
Negative	196 (85.2%)
Isolated tumor cells	6 (2.6%)
Micrometastasis	8 (3.5%)
Macrometastasis	20 (8.7%)

BMI = body mass index. FIGO= International Federation of Gynaecology and Obstetrics.

**Table 2 jcm-09-01999-t002:** Comparison of uterine histopathological risk factors by nodal metastasis size. Univariate and multivariate analyses.

	Negative Lymph Nodes	Low Volume Metastasis	Macrometastasis	Univariate	Multivariate
	N(%)	N(%)	N(%)	*p*-Value	*p*-Value
Myometrial invasion					
<50%	130 (66.3)	7 (50)	3 (15)	*p* < 0.001	*p* = 0.032
≥50%, cervical stromal invasion	66 (33.7)	7 (50)	17(85)		
Histology					
Endometrioid	164 (83.7)	12 (85.7)	15 (75)	*p* = 0.593	*p* = 0.593
Non-endometrioid	32 (16.3)	2 (14.3)	5 (25)		
Grade					
G1	104 (53.1)	6 (42.9)	5 (26.3)	*p* = 0.117	*p* = 0.990
G2	36 (18.4)	5 (35.7)	5 (26.3)		*p* = 0.887
G3	56 (28.6)	3 (21.4)	9 (47.4)		*p* = 0.941
Lymphovascular invasion					
No	152 (78.4)	6 (42.9)	6 (31.6)	*p* < 0.001	*p* = 0.054
Yes	42 (21.6)	8 (57.1)	13 (68.4)		

**Table 3 jcm-09-01999-t003:** Adjuvant therapy by nodal status. VBT: vaginal brachytherapy; EBRT: external beam radiotherapy; CT: chemotherapy.

	Node-Negative	LVM	Macrometastasis	*p*
**Follow up or VBT**	74.2% (135)	28.6% (4)	5.6% (1)	*p* < 0.001
**EBRT ± BT**	13.7% (25)	28.6% (4)	11.1% (2)	*p* < 0.001
**CT ± EBRT**	12.1% (22)	42.9% (6)	83.3% (15)	*p* < 0.001

**Table 4 jcm-09-01999-t004:** Characteristics of patients with LVM nodal status.

	Type LVM	Stage	Histology	Grade	Lymph-Vascular Invasion	Adjuvant Treatment	Recurrence
Patient 1	ITC	IB	Endometrioid	3	+	CT ± EBRT	None
Patient 2	ITC	IB	Carcinosarcoma	3	+	CT ± EBRT	Yes
Patient 3	ITC	IA	Endometrioid	2	-	None	Yes
Patient 4	ITC	IA	Endometrioid	1	-	EBRT ± BT	None
Patient 5	ITC	IB	Endometrioid	2	+	CT ± EBRT	None
Patient 6	ITC	IA	Endometrioid	1	-	None	None
Patient 7	MIC	IA	Endometrioid	1	+	VBT	None
Patient 8	MIC	IB	Endometrioid	1	+	EBRT ± BT	Yes
Patient 9	MIC	IB	Endometrioid	1	+	EBRT ± BT	None
Patient 10	MIC	IA	Endometrioid	1	+	CT ± EBRT	None
Patient 11	MIC	IB	Endometrioid	2	+	EBRT ± BT	Yes
Patient 12	MIC	IA	Endometrioid	2	-	None	None
Patient 13	MIC	IA	Endometrioid	3	-	CT ± EBRT	None
Patient 14	MIC	IB	Endometrioid	1	-	CT ± EBRT	None

MIC = micrometastasis; ITCs: isolated tumor cells; VBT: vaginal brachytherapy; EBRT: external beam radiotherapy; CT: chemotherapy.

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
