# Peer review of "Low-Volume Nodal Metastasis in Endometrial Cancer: Risk Factors and Prognostic Significance"

_jcm, 2020, doi:10.3390/jcm9061999_

Round 1

Reviewer 1 Report

SUMMARY

The authors have presented an interesting topic regarding the prognostic impact of low-volume metastases identified by use of sentinel lymph node (SLN) techniques. They revealed a significantly poorer overall and recurrence-free survival for patients with macrometastases, and no significant differences in survival between patients with low-volume metastases as compared to negative lymph nodes. As an increasing number of hospitals are introducing SLN biopsies as part of standard surgical procedure, it is highly relevant to assess the prognostic impact of the small lymph node metastases that are identified through SLN techniques. In that sense, these findings may contribute to the development of new guidelines for patients with low-volume lymph node metastases. However, this finding is not novel. This has already been investigated in larger patient cohorts (e.g. St. Clair and Plante et al).

BROAD COMMENTS

In general, the language in this paper draft need be improved. It needs substantial revision; particularly the results and discussion part. The method lacks description of which test is used to calculate p-value for Kaplan Meier analyses. The results are not presented in a clear fashion and contain too much details in text on actual numbers of patients/percentages. This can be better communicated by referring to tables and commenting on which associations that are significant, and in which direction. The language in the discussion part is poor with several grammatical errors, and it is challenging to follow the author’s reasoning. Further, the discussion enlightens the fact that these are not novel findings.

SPECIFIC COMMENTS

Abstract:

The language is OK. The colon in line 18 looks to be misplaced. It appears to be a typo in line 26. Please change from that to than. Please indicate what time point the survival refers to.

Introduction:

Why is US epidemiological statistics used, and not global? Why are only US guidelines referred to?

Methods:

Please insert what test is used to calculate p-value for Kaplan Meier analyses.

Results:

Actual patient numbers with percentages should not be presented in the text to such great extent. Instead, significant p-values should be commented as text with comments on which histologic feature is enriched in which group.

Figure 1 and 2: Kaplan-Meier plots: It is not clear if the percentages used to distinguish differences in survival between groups are at three, five or perhaps eight years of survival. The Kaplan-Meier curves are insufficiently presented. Formatting is needed. P-values should be presented in the plots.

Discussion:

Please improve the way you refer to other studies. Example of poor referring: Line 176-178

Please go through how you refer to cohorts. Example of poor writing: Line 196: ‘In our results, cohort of macrometastasis..’ Please substitute with: ‘In this study, patients with macrometastases…’

Please go through how you present clinical data. Example of poor writing: Line 242-244: Please do not write ‘depth of myometrial invasion ≥50%, instead write ≥50% myometrial invasion. Also, do not refer to clinicopathological characteristics as they. Rephrase sentence. Patients could be referred to as they, not clin.path characteristics.

Reviewer 2 Report

Specific comments:

Ganglionar should be replaced by “lymph node” throughout the manuscript since ganglionar is not a common English term for lymph node and is confusing for the reader.

In general the text of the results section is too extensive and detailed considering that it is supported by the very informative Tables and Figures. Please summarize the most important findings and refer to the Tables for more details in order to avoid extensive overlap.

Line 106: “diagnosed of not disseminated» Please rephrase and clarify what this meant? Putative stage 1 and 2?

Please use dots instead of comma for decimal separator when giving numbers (e.g. 85.2% instead of 85,2%) throughout the manuscript.

Line 127: Please give separate subnumbers for the nine cases with LVM: How many of these were diagnosed with ITC and how many with MIC?

Line 128-9: Change text to: ”Among the positive SLNs nodes (34/314; 10.8%), ….”

Figures should be made high-quality using correct numbers (using dots instead of commas as decimal separators) and more professional labeling and headings. Please provide patient numbers and events for each category in the survival plots and p-values given in the plots.

Figure 1: Is micrometastasis correct or should it be macrometastasis? I assume it is macrometastases?

Table 4. Order patients so that you start with all ITC follows by all MIC.

Discussion Lines 203-204: Poor English – please rephrase.

Lines 228-230. Please give the p-value for this comparison. If the p-value is close to 0.05 you can report it as a tendency/trend, although not significant. If the p-value is close to 1.0, it is not justified to make a point that there is a tendency that you were not able to show (due to small sample size). Anyhow I think you should be very careful about drawing conclusions on this considering the low patient number with ITC/micrometastases.

Lines 234-236: What does the p-value refer to here? Please rephrase/clarify.

251-271: Check by native English speaking expert to improve clarity.

Reviewer 3 Report

García Pineda et al have evaluated the oncological outcomes of patients with low-volume metastasis (LVM) compared to women with macrometastasis and women with negative nodes in endometrial cancer. A total of 270 patients were operated in the study period. After a median follow-up of 60 months, women with macrometastasis showed a significantly shorter progression free survival (PFS) and overall survival (OS) compared to LVM and node negative patients (61.1% vs 71.4% vs 83.2% and 50.0% vs 78.6% vs 81.5%, respectively). Women with LVM had a moderate non-significant decrease in PFS compared to node negative patients. In conclusion, women with LVM did not show worse oncological outcomes that node negative patients although a slight decrease in progression free survival was observed.

The claims are properly placed in the context of the previous literature. The experimental data support the claims. The manuscript is written clearly enough that most of it is understandable to non-specialists. The authors have provided adequate proof for their claims, without overselling them. The authors have treated the previous literature fairly. The paper offers enough details of methodology so that the experiments could be reproduced.

Minor revisions

In English, periods (".") are used and not commas (",") in percentages and p-values.

Page 1, line 16, "On final pathology, 196 (85,2%)" => "On final pathology, 196 (85.2%)"

Page 1, line 22, "compared to LVM and node negative patients (61,1% vs 71,4% vs 83,2%, respectively; p=0,018)" => "compared to LVM and node negative patients (61.1% vs 71.4% vs 83.2%, respectively; p=0.018)"

This has to be corrected in the whole manuscript.

Page 6, Figure 1, Progression-Free Survival (PFS) by nodal status. The label for "Nodal status" has to be corrected from "Micrometastasis" to "Macrometastasis" (as in Figure 2).

Round 2

Reviewer 1 Report

General comments:

The manuscript has been improved and the conclusions are more suitable to the results and the novelty of this study. The results part is sufficiently improved. The discussion still needs major improvements. Please be consistent with how you refer to groups/cohorts. Please go over how you report on clinical data concerning patients. I have included several specific comments on the discussion that I believe will help resolve some of these issues. Please also be consistent with how you write >50% myometrial invasion. Please also note that numbers smaller than or equal to ten normally are spelled out (e.g. two instead of 2).

The figures are still not professionally displayed. I recommend you to use an editor program like adobe illustrator to make the KM plots publishable.

Specific comments:

Experimental section:

Line 76: Please change from serous papillary to serous (according to my knowledge, this is currently the official naming of the subtype).

Discussion:

Line 254-256: Please rephrase: ‘the patients treated between (…) did not include it in their nodal staging. If I understand this correct, what you mean is: for the patients treated between (…), SLNB was not routinely performed,’

Line 260: Please rephrase and add something about what kind of evidence you are referring to.

Line 274: Please rephrase the sentence, specifically: ‘showed also’ and ‘the 47%’ and comma before respectively.

Line 281: Please rephrase: ‘LVM were supposed 54%’

Line 284: Please rephrase to LVM were detected in 14 patients (41.2%).

Line 284-285: I believe what you mean is: ‘in accordance with published literature’. And please insert one or more references.

Line 286: Please rephrase ‘unclear nowadays’ to ‘is still unclear’

Line 290: Please rephrase sentence to ‘Nevertheless, within the LVM cohort, 3 (21.4%) patients with ITC received chemotherapy ±EBRT and 1 (7.14%) patient received EBRT±BT based on high risk uterine features.’

Line 292: Please rephrase similarly to the suggestion for line 290: ‘within the micrometastasis cohort, 3 (21.4%) patients received (…) ’

Line 242: Please rephrase to (or similar): Despite the differences in adjuvant treatment administered to patients according to nodal status,

Line 296: Please rephrase to: ‘The OS within the macrometastasis cohort’

Line 298: Please rephrase to: ‘in accordance with published literature’ and insert one of more references

Line 299: please remove ‘with’

Line 299/300: Please insert a ‘the’ before ITC cohort.

Line 301: Please rephrase to: patients with macrometastasis.

Line 301-302: Please rephrase to: Further, patients with macrometastasis had a significantly poorer survival than patients with ITC (?) (58.6% vs 95.5%).

Line 305: Please rephrase to: Clair et al reported similar results

Line 308: Please rephrase to: ‘between adjuvant treatment administered to the three different groups.

Line 309: Please insert ‘the’ before negative nodes group and ‘it’ before LVM.

Line 314: There is one extra space between those and of

Line 326: Please rephrase to: Within the LVM cohort, 4 patients experienced recurrence, including 2 cases with ITC and 2 with micrometastasis.

Line 329: please insert: reported or published before literature.

Line 334: Please rephrase to: ‘ however not statistically significant’.

Line 337: Please insert: ‘groups’ or ‘cohorts’ in the end of the sentence.

Line 341-345: I do not understand the reasoning. At first in the sentence, you claim that Clinton’s study is in consistence with yours, yet as far as I can see, you claim the opposite when reporting on your results (instead in our LVM cohort, (…) ).

Line 351: Please rephrase to: In this study, we report no differences (…)

Line 357: Please rephrase to: However, a higher recurrence rate was observed in the LVM cohort as compared to the node negative patients;

Line 364: Please insert: ‘the’ before prognostic

Line 366: Please use ‘administered’ instead of ‘performed’.

Conclusions:

General comment: I would prefer to start the conclusion by ‘In conclusion’ or ‘We here present’ or ‘In this study’

Line 374: Please rephrase to: In this study, patients with LVM had significantly better survival than patients with macrometastasis.

Line 375-376: Please rephrase sentence.

Author Response

This manuscript is a resubmission of an earlier submission. The following is a list of the peer review reports and author responses from that submission.